# Cross-Sectional 4D-Printing: Upscaling Self-Shaping Structures with Differentiated Material Properties Inspired by the Large-Flowered Butterwort (*Pinguicula grandiflora*)

**DOI:** 10.3390/biomimetics8020233

**Published:** 2023-06-02

**Authors:** Ekin Sila Sahin, Tiffany Cheng, Dylan Wood, Yasaman Tahouni, Simon Poppinga, Marc Thielen, Thomas Speck, Achim Menges

**Affiliations:** 1Institute for Computational Design and Construction (ICD), University of Stuttgart, 70174 Stuttgart, Germany; dylan.wood@icd.uni-stuttgart.de (D.W.); yasaman.tahouni@icd.uni-stuttgart.de (Y.T.);; 2Cluster of Excellence IntCDC, University of Stuttgart, 70174 Stuttgart, Germany; 3Botanical Garden, Department of Biology, Technical University of Darmstadt, 64287 Darmstadt, Germany; simon.poppinga@tu-darmstadt.de; 4Plant Biomechanics Group, Botanic Garden, University of Freiburg, 79110 Freiburg, Germany; marc.thielen@biologie.uni-freiburg.de (M.T.); thomas.speck@biologie.uni-freiburg.de (T.S.); 5Cluster of Excellence *liv*MatS @ FIT—Freiburg Center for Interactive Materials and Bioinspired Technologies, University of Freiburg, 79110 Freiburg, Germany

**Keywords:** additive manufacturing, shape change, material programming, adaptive structures, mechanical metamaterials, functional materials, carnivorous plants, bioinspiration

## Abstract

Extrusion-based 4D-printing, which is an emerging field within additive manufacturing, has enabled the technical transfer of bioinspired self-shaping mechanisms by emulating the functional morphology of motile plant structures (e.g., leaves, petals, capsules). However, restricted by the layer-by-layer extrusion process, much of the resulting works are simplified abstractions of the pinecone scale’s bilayer structure. This paper presents a new method of 4D-printing by rotating the printed axis of the bilayers, which enables the design and fabrication of self-shaping monomaterial systems in cross sections. This research introduces a computational workflow for programming, simulating, and 4D-printing differentiated cross sections with multilayered mechanical properties. Taking inspiration from the large-flowered butterwort (*Pinguicula grandiflora*), which shows the formation of depressions on its trap leaves upon contact with prey, we investigate the depression formation of bioinspired 4D-printed test structures by varying each depth layer. Cross-sectional 4D-printing expands the design space of bioinspired bilayer mechanisms beyond the XY plane, allows more control in tuning their self-shaping properties, and paves the way toward large-scale 4D-printed structures with high-resolution programmability.

## 1. Introduction

Four-dimensional printing (4D-printing) is an emerging additive manufacturing technique that produces material systems that change shape over time in response to environmental stimuli [1,2]. Using this technique, material systems can be printed in their flat state and actuated into their programmed shape upon exposure to external stimuli [3,4]. The actuation can be triggered by various stimuli, such as temperature [5,6], moisture [4,7], humidity [8], or magnetic fields [9]. The self-shaping process is generally tuned by the mesoscale structuring of stimuli-responsive materials. For instance, one can tailor the internal composition of material systems [10] and create heterogenous mesostructures using fused deposition modeling (FDM) [11,12]. The control of extrusion paths results in a tailored anisotropy within the structure, which helps to achieve the desired shape and response to external triggers. In essence, mesostructures act as coded programs that dictate self-shaping.

Much of the existing literature on self-shaping is inspired by plant motions [13,14,15,16], which are driven by slow water displacement processes between cells and tissues, including reversible turgor changes, hygroscopic reactions or irreversible growth, and/or the rapid release of elastic energy (prestress) previously stored in the motile structures [17]. Plants are able to perform a variety of motions based on deformation processes, such as bending [18], twisting [19], coiling [20], snap through [21], and explosive bursting [22].

Hygroscopic actuation principles are based on their functionally graded internal structures, which include fibrous and anisotropic cellulose microfibrils as well as hygroscopic tissues and cells. For example, pinecone scales are composed of sclereid and sclerenchyma tissue layers arranged in opposing orientations with differential swelling and shrinking properties within the bilayer structure [23,24], which produces bending in the pinecone scale and enables the release of seeds. Pinecone scales have been one of the chief role models for bioinspired 4D-printed structures, and considerable research has focused on the production of surface-based composite bilayers [25,26,27].

In reality, biological materials systems are far more complex than a simple bilayer. A recent study proved that the pinecone scale is in fact an intricate composite structure consisting of various functionally significant layers [28]. Contrary to current applications of 4D-printing, biological tissues are comprised of different material layers with graded mechanical properties across the scales. In practice, FDM 4D-printing typically operates in a 2.5D manner and builds structures layer by layer [29,30,31,32]. While extrusion paths can be controlled to tailor mechanical properties, it is limited mainly to the XY plane. Although modulating properties in the Z axis is possible, the increased depth and thickness result in shape changes with slower actuation and less bending curvature [30]. To amplify the bending curvature, bilayers are often printed with multiple materials that are difficult to recycle. Furthermore, exploring the design space of 4D-printing has often been an empirical study. Even though research on the simulation aspects has been conducted [33,34], the focus has been on shape deformation, which lacks connection to the production of 4D-printing. Due to these limitations, 4D-printed bilayers have been restricted in size and potential for biomimetic transfer.

### 1.1. Bioinspiration: Depression Formation in Large-Flowered Butterwort (Pinguicula grandiflora)

One promising yet little explored example of complex shape formation in nature is the peculiar movement of the leaves of the carnivorous large-flowered butterwort (*Pinguicula grandiflora*). Large-flowered butterworts have motile trap leaves that attract, capture, retain, kill, and digest small arthropod prey in nutrient-poor habitats [35,36]. When mechanically and/or chemically stimulated by prey, the large-flowered butterwort responds by forming a depression in the leaf lamina around the prey [37,38]. The localized forming of a depression on an otherwise laminar structure (i.e., a leaf) is a very specific mechanical response that can be clearly differentiated from other, more extensive and large-scale leaf motions such as the bending of carnivorous sundew (*Drosera*) leaves [39] and leaf rolling in cold-hardy *Rhododendron* [40]. Although not yet understood from a structural–mechanical point of view, the basis of the large-flowered butterwort’s depression formation is caused by cell- and tissue-specific changes as well as the complex interactions between them in the depth layers of the leaf lamina. A better understanding of this tailored and locally confined movement phenomenon, which has a high biomimetic potential, would allow the development of novel systems and materials capable of responding to specific stimuli in a localized manner [41].

### 1.2. Proposed Concept: Cross-Sectional 4D-Printing

We propose what we term the cross-sectional 4D-printing (CS4DP) method of tailoring functional properties across the depth of “bilayers” by printing their cross sections in a rotated axis. In typical 4D-printing processes, two material layers with different hygroscopic properties are printed sequentially to produce a bilayer structure. This standard process results in solid and double-layered cross sections, which limits their shape change on larger scales. On the other hand, CS4DP structures are constructed by building their cross sections in the XY-plane and extruding them along the Z axis to the desired height (Figure 1a). This allows for the precise arrangement of regional depth layers (RDLs) with differentiated hygroscopic and mechanical properties within the cross section of the structure (Figure 1b). As a result, CS4DP allows for the production of material systems with higher intricacy and porosity (Figure 1c), with significant shape changes at increased scales, even as a monomaterial system.

Our work is inspired by the cellular transformations of the large-flowered butterwort (*Pinguicula grandiflora*) and the resulting shape change in its cross section (Figure 2a). In a related biomimetic study, La Porta et al. [39] investigated a metamaterial bilayer that bends when subjected to orthogonal compression. While researchers have primarily concentrated on mechanically induced shape change, our study examines the self-shaping effect of hygroscopic functional patterns in CS4DP structures.

Through physical experiments on 4D-printed specimens, we determined the effect of functional pattern parameters on hygroscopic and mechanical properties (Figure 2b,c). We then correlated the functional patterns and their hygroscopic properties with an abstracted digital voxel-based model (Figure 2d), which allowed the depth layers of 4D-printed specimens to be programmed with varying hygroscopic properties and their shape changes to be simulated. This aids in the design of functionally graded material systems with desired shape changes (Figure 2e,f). Finally, we investigated the depression formation in the large-flowered butterwort by means of a comparative functional–morphological analysis and then extrapolated its functional principles by using CS4DP with differentiated mechanical and hygroscopic properties.

## 2. Materials and Methods

### 2.1. Material, Machine, and Software Workflow for 4D-Printing

A commercial FDM 3D printer (FELIX Tec 4, FELIXprinters, Ijsselstein, Netherlands) with a printable bed size of 240 × 205 mm was employed for the fabrication process. To create a monomaterial system, only one of the extruders with a 0.35 mm nozzle was used to extrude a moisture-responsive filament (Laywood Meta 5, LayFilaments, Cologne, Germany) with a diameter of 1.75 mm. The material showed significant dimensional changes depending on the relative humidity (RH) level of the environment or the amount of water absorption [42]. The print bed temperature and the nozzle temperature were kept constant at 55 °C and 195 °C, respectively.

The deposition of the material was controlled via a custom Gcode, which dictates the 3D printer’s tool path and extrusion parameters. Within a graphical algorithm editor (Grasshopper 3D) as a part of a commercial computer-aided design software (Rhinoceros 3D), printing tool paths and extrusion parameters were defined, sequenced as a single crossing and multilayered line, visualized (to ensure a smooth printing process), and translated into Gcode [43]. The speed of printing was kept constant at 1600 mm/s to ensure a consistent print quality and shape programmability [44]. Unless otherwise indicated, all 4D-printed functional patterns were extruded with 0.05 mm of filament per linear mm.

### 2.2. Model Representation of Differentiated Cross Sections

Functional patterns, which control the mechanical and hygroscopic properties at a high resolution [45], were utilized for the proposed 4D-printing technique. The base designs were generated from a structured grid of cells with variable resolution (via the size of individual cells), wall thickness (via the printing flow rate), and wall geometries (via the printing tool path) (Figure 3). Functionally discrete or graded regions could also be designed and arranged in a cross-sectional structure.

To model the behavior of cross-sectional structures with diversified properties, we developed a voxel-based model derived from constraints-based projection modeling [46] implemented using a live physics engine (Kangaroo 2) in the same design and fabrication environment of CS4DP. The voxel sizes were defined considering the cell resolution of each RDL. To prevent the delamination of the RDLs in the physics engine, the voxel sizes were slightly adjusted in the X direction so that the voxels’ anchor points were matched in the digital model. Regions of the voxels were assigned different deformation qualities, and the deformation of each voxel was simplified by scaling them along the X and Y directions. Through physical tests, directional shrinkage data from the 4D-printed functional patterns could be correlated to the voxel-based model in order to simulate the shape deformations more accurately.

### 2.3. Experiments and Measurement of 4D-Printed Specimens

Four-dimensional printed test specimens were measured in size and weight in three different states: the initial state (dry, after printing), the wet state (after 24 h water submergence), and the dry state (after removal from water and placement in 30% RH on a low friction surface for 48 h). For each experiment, three test samples were printed and measured. The measurements obtained from each sample were then averaged.

For each state, the width and length of the samples were measured from three different points by using an electronic pocket vernier caliper with a 150 mm measuring range and 0.01 mm of readability (ORION, HAHN+KOLB Group, Ludwigsburg, Germany). For specimens longer than the caliper, a millimetric ruler was used (Soennecken, Soennecken eG, Overath, Germany). The average of those values was used to calculate the amount of shape deformation as a percentage. In order to identify the change in the moisture content, the specimens were weighed by using a digital precision scale, which measures weights with a 500 g measuring range and 0.01 g readability (Foraco, Lotus NL B.V., The Hague, Netherlands).

To evaluate the mechanical performance of the specimens, pull tests were conducted on the 4D-printed samples by using a digital spring scale (Rhorawill, Guangzhou Weiheng Electronic Technology Co., Ltd., Guangdong, China). The tests involved incrementally applying force to deform the printed structures until failure. For each functional pattern, three samples were tested, and the corresponding load-displacement data were recorded.

### 2.4. Plant Material, Feeding, and Functional–Morphological Analysis

*Pinguicula grandiflora* was originally purchased from a commercial supplier (Carnivores & More, Merzig, Germany) and were cultivated in a temperate greenhouse of the Botanical Garden of the University of Freiburg according to its requirements (Figure 4a). The formation of the depressions was induced by carefully placing live fruit flies (*Drosophila melanogaster*) with forceps on the central regions near the midribs of the fully developed leaves. The motions were recorded by using an MZ-902 USB camera (OOWL Tech Ltd., Hong Kong, China).

Trap leaves with a depression resulting from stimulations by prey were embedded in Technovit7100 (standard procedure) (Heraeus Kulzer GmbH, Wehrheim, Germany). A custom-made rotating microtome was used to produce 5 μm thick semithin leaf cross sections that showed the depression on one half of the leaf, the midrib, and the other half of the leaf without a depression. Toluidine blue staining was applied (infiltration for 1 min in toluidine, 1 min washing with deionized water) and the microscopy slides were sealed with Entellan (Merck KgaA, Darmstadt, Germany). Sections were investigated with a BX61 light microscope equipped with a DP71 digital camera (Olympus Corp., Tokyo, Japan). The leaf dimensions and cell sizes were measured comparatively by using Fiji/Image J [47].

## 3. Results

### 3.1. Analysis of the Large-Flowered Butterwort’s Deformations

Feeding plants with fruit flies resulted in slow leaf margin rolling and/or depression formation within ~24 h (Figure 4b,c) (Appendix A). The semithin section with toluidine blue staining of a triggered trap leaf clearly showed the midrib with conducting strands, which divided the leaf into left and right halves (Figure 4d). While the right half of the leaf shown clearly responded to the presence of a prey animal with the formation of a depression (L_stimulated_), the left side (which was not stimulated with prey) did not (L_non-stimulated_).

L_stimulated_ showed clear differences in tissue and cellular dimensions when compared to L_non-stimulated_. The overall mean ± SD leaf thickness was 225 ± 12 µm (*n* = 10) for L_non-stimulated_ and 199 ± 24 µm (*n* = 11) for L_stimulated_. The adaxial epidermal cells analyzed in L_stimulated_ in the vicinity of the depression were 37 ± 13 µm (*n* = 69) wide and 15 ± 5 µm (*n* = 69) high, as seen in the cross section, and those for L_non-stimulated_ in a comparable area (i.e., mirrored about the midrib) were 35 ± 14 µm (*n* = 76) wide and 25 ± 10 µm (*n* = 76) high. Generally, the mean thickness of the adaxial epidermis in L_stimulated_ in the vicinity of the depression accounts for only ~60% of the mean thickness of the epidermal layer in L_non-stimulated_ in a comparable area. Whereas the abaxial epidermis in the L_non-stimulated_ region (comparable in size to the stimulated one on the other leaf side) is characterized by cells with average widths of 41 ± 26 µm (*n* = 71) and heights of 34 ± 12 µm (*n* = 72), cells in the L_stimulated_ region are 34 ± 17 µm (*n* = 96) wide and 41 ± 15 µm (*n* = 99) high. In summary, whereas the mean abaxial epidermal cell widths in L_stimulated_ are ~17% smaller, the mean heights are ~21% greater than in L_non-stimulated_.

Based on the above-described anatomical analyses, the following actuation scenario for depression formation in *P. grandiflora* can be hypothesized (Figure 4e,f). In agreement with Heslop-Harrison and Knox [38], the shrinkage of the adaxial epidermis was observed. Additionally, the abaxial epidermal cells became thicker (especially higher, as seen in the cross section). Due to the thickness changes of the entire leaf, mesophyll cellular changes can also be assumed but remain to be investigated. The cellular dimensional changes were likely responsible for the overall leaf deformation and could be attributed to the complex water displacement processes between the tissues.

### 3.2. Evaluation of the 4D-Printed Specimens

To test the aforementioned hypothesis about the large-flowered butterwort’s shape change, CS4DP was used to help elucidate the functional principle behind the large-flowered butterwort’s depression formation based on the observed degrees of cell shrinkage. A first test series was conducted to gain the shrinkage, ultimate load, and deformation values of the uniform functional patterns, and a second test series was conducted to assess varied arrangements of cross sections with distinct depth layers.

#### 3.2.1. Uniform Functional Patterns

We 4D printed test specimens with an overall boundary of 100 × 100 × 5 mm. Each specimen was printed with a total of 20 layers and at a layer offset of 0.25 mm. Variations in the following parameters were changed uniformly within the same boundary and compared: cell resolution, wall thickness, and wall geometry. Each parameter was printed in three different degrees: high, medium, and low. Anisotropy was also tested by varying the parameters in X and Y directions. The influence of these parameters on global shape transformation and deformation under incremental loading were then evaluated (Figure 5).

Regarding shape transformation, a contrasting relationship between the swelling and shrinkage data was observed when evaluating all the functional pattern tests with three samples. Specifically, the functional patterns that exhibited the highest levels of shrinkage often displayed the least amount of swelling after being submerged in water and vice versa.

Focusing on the parameter tests, changes in the cell resolution (Figure 5a) did not show a clear trend in shape deformation. However, it was observed that the functional patterns with the highest resolution (the smallest cell size, more material deposition overall) achieved higher shrinkage and thus shape change. Unlike the cell resolution, changes in the wall geometry (Figure 5b) indicated a clear trend. Two types of geometries were tested: a sinusoidal curve and a squiggle curve. The squiggle curve was tested in anisotropic patterns to assess the effects of added curve lengths, whereas the sinusoidal curve was tested in both isotropic and anisotropic patterns. It should be noted that with higher curve amplitudes (and thus more added curve lengths), the flexibility of the specimen increased and became more prone to being shaped by hand, especially during removal from water. Consequently, even though specimens with a high curve amplitude in their wall geometry were produced, they were not evaluated due to their malleability in the wet state. In general, with the higher curve amplitude in the wall geometry, less shrinking and more swelling were observed. This was clearly shown in the anisotropic patterns, where there was more shrinking in the straight walls and more swelling in the curved walls. Likewise, variations in the wall thickness (Figure 5c) demonstrated a pronounced correlation with its hygroscopic properties. Decreasing the wall thickness (reducing the flow rate and thus the material deposition) resulted in increased shrinkage, while the increased wall thickness caused the structure to change shape less. On the other hand, as the wall thickness increased, so did the amount of swelling. Overall, the largest shrinkage was observed in the functional patterns with low wall thickness, low curve amplitude, and high cell resolution.

With a primary focus on hygroscopic evaluation, we also conducted mechanical performance assessments of the functional patterns. The average ultimate load and deformation data were recorded from three samples in their dry state. The results of the mechanical pull tests indicated that the printed structures with a higher shape change capability can also bear higher ultimate loads. For example, functional patterns with anisotropic resolution exhibited notable shrinkage while withstanding an average force of 45.3 N. In contrast, the pattern with low cell resolution could only sustain a maximum load of 11.3 N.

Similarly, the wall geometry patterns exhibited a parallel trend. To exemplify, the functional pattern with a lower sinusoidal amplitude (added curve length of +1.64 mm in both directions) shrunk more than the one with the higher sinusoidal amplitude (added curve length of +4.2 mm), as mentioned (Figure 5b). As a result of their mechanical assessment under incremental loading, sinusoidal wall geometry with lower amplitude exhibited an ultimate force capacity of 37.9 N, whereas the one with a higher amplitude could only withstand 24.9 N. It is important to note, however, that the latter demonstrated an elongation of ≈37 mm before failure, while the former only had an elongation of ≈17 mm.

Nevertheless, the results of the mechanical pull tests revealed that the functional patterns with a higher shape change capability cannot always be associated with higher ultimate loads. As an example, the functional pattern with low wall thickness demonstrated a high shape change (−1.7% in both directions) while bearing a load of 22.9 N. Additionally, as the wall thickness (flow rate or E value) increased, the functional pattern exhibited a decrease in shape change while showing an increased ability to bear higher loads.

Based on the insights gained from the mechanical tests and hygroscopic experiments, cross-sectional patterns were designed to achieve the desired shape change. This was achieved by incorporating the percentage of dimensional shrinking and swelling observed for each functional pattern into a voxel-based model. Through simulations of shape deformation, the voxel-based model informed the design of cross sections with distinct depth layers. The trends observed between the different degrees of each parameter helped to estimate the hygroscopic behavior of untested functional patterns in the design of cross sections with distinct depth layers.

#### 3.2.2. Cross Sections with Distinct Depth Layers

Test specimens were designed with multiple distinct regional depth layers (RDLs). Based on the analysis of the large-flowered butterwort’s deformation, the hypothesis about the cellular interactions and properties between its epidermis and mesophyll depth layers were translated into the representative functional pattern for each RDL. Before 4D-printing, we first simulated the shape change of the designed specimens through the voxel-based model, which also provided feedback on the RDL’s programmed properties for an additional optional iterative design cycle. We then 4D printed test specimens with an overall boundary of 220 × 34 × 5 mm. Each specimen was printed with a total of 20 layers and at a layer offset of 0.25 mm, as in the uniform functional pattern tests. After the water submergence and subsequent drying, we compared the bending curvature resulting from their shape deformations (Figure 6).

We designed a cross-sectional bilayer specimen (Figure 6a) based on the thinning of the upper epidermis and the thickening of the lower epidermis via an RDL 1 programmed with anisotropic shrinking and an RDL 2 programmed with an expandable functional pattern. A curvature of 0.0011 mm^−1^ was predicted in the voxel-based model, while the results measured an actual curvature of about 0.0019 mm^−1^.

Next, we designed a cross-sectional tetralayer specimen (Figure 6b) that mimicked the upper epidermis (RDL 1), upper mesophyll (RDL 2), lower mesophyll (RDL 3), and lower epidermis (RDL 4). RDL 1 was programmed to decrease in thickness; RDL 2 was programmed with an unshrinkable and pullable functional pattern; RDL 3 was programmed to undergo a dramatic shrinkage; and RDL 4 was programmed to increase in thickness. A curvature of 0.0009 mm^−1^ was predicted in the voxel-based model, while the results measured an actual curvature of about 0.0010 mm^−1^.

A higher curvature was observed in the cross-sectional bilayer than in the cross-sectional tetralayer. In order to demonstrate our system’s scalability, we produced another specimen of the cross-sectional bilayer design with an increase in the printed height (from 5 to 9 mm), which proved that a similar curvature could be maintained from the upscaling. We also produced another specimen of the cross-sectional tetralayer design with an increase in the cross-sectional depth (from 33. 1 to 58.7 mm), which exhibited an even higher curvature of approximately 0.0012 mm^−1^.

## 4. Discussion

In contrast to the standard bilayer composite system (containing two thin surfaces of contrasting materials) printed on the XY plane, the developed CS4DP technique results in differentiated and layered structures with varied properties in the cross section. As a result, porous and upscaled self-shaping structures (from 1 to 58.7 mm or more) can be produced, even as a monomaterial system and by using commercially available materials and desktop printers. CS4DP shifts the traditional way we think about bilayer systems and self-shaping architectures. From a methodological perspective, the rotation of the bilayer printing axis redefines not only the printing process parameters but also the design parameters. Using this technique, the mesostructure required for shape transformation is generated by functional patterns, which can be varied in depth through layering within the cross section.

To design and produce the cross-sectional structures, an understanding of the relationship between the functional pattern parameters and their resultant hygroscopic and mechanical properties is necessary; it is also challenging to predict how the shrinkage of each depth layer in the cross-sectional structure will affect the total shape transformation. Hence, we produced, triggered, and measured 4D-printed test specimens of uniform functional patterns with the shrinkage data input into a digital voxel-based model. By then abstracting functional patterns as voxels with programmable hygroscopic properties, we provided a simple, intuitive, and fast method to model and simulate different design iterations of RDLs before producing the cross-sectional structures.

This workflow can then also be used to examine the actuation principles behind the large-flowered butterwort’s depression formation. We test the hypothesis that the dramatic shrinkage of the mesophyll, combined with the thinning (shrinking) of the adaxial (upper) epidermis and thickening (expanding) of the abaxial (lower) epidermis is responsible for the initiation of the depression formation. This approach can allow other scientists to verify and improve their hypotheses on the biomechanics of new plant role models, which offers further insights from engineering to biology through reverse biomimetics [48].

Our proposed methodology still requires further empirical studies to precisely estimate not only the effect of the functional pattern parameters but also their combinations. In order to simulate shape changes more accurately, the developed voxel-based method can be extended to interface with Finite Element Modeling (FEM) software. The FEM simulation can then be utilized to quantify the stresses generated in different RDLs, which will be crucial for the upscaling process. Accordingly, an in-depth mechanical characterization of the material system in addition to an FEM analysis of the structures will benefit future work. FEM can also be incorporated into methods that use self-organization principles such as agent-based modeling (ABM) [49]. This would enable the introduction of other cell properties and behaviors beyond dimensional changes. By learning from the resultant shape changes, another promising future development would be inverse design.

In addition, the long-term durability of the structures and possible delamination due to cyclic actuation should be carefully considered. The occurrence of delamination has been minimal in the presented study, thanks to the good mechanical properties in the bending plane due to the CS4DP process as well as the good adhesion between the layers due to the monomaterial system. However, it should be noted that even in monomaterial systems, delamination can occur if the printing parameters (layer height and material flow rate) are not calibrated.

Although the focus of this research has been on the singular cross section, multiple cross sections could be interpolated into a greater surface structure. In this way, one can design several differentiated sections and create a whole structure with double curvature shape deformation. Cross-sectional structures could also be integrated with conventional bilayers, which enables the design of structures with programmed shape deformations in the XY, XZ, and YZ planes. Furthermore, the applications of 4D-printing can be expanded on a large scale [50] by combining cross-sectional structures with other structural and hygroscopic material systems, such as hygroscopic wood elements. Upscaling 4D-printed structures requires further considerations in terms of their structural integrity. For example, compared to plant structures, printed structures mostly lack the tensile strength in the Z direction. The structural capacity of the printed structures should be investigated in upscaling or more complex deformations.

## 5. Conclusions

This research is based on a cross-disciplinary concept inspired by a carnivorous plant named the large-flowered butterwort (*Pinguicula grandiflora*). It introduces a new bioinspired additive manufacturing technique called cross-sectional 4D-printing (CS4DP), which transfers the deformation principles from motile plants to material systems in cross sections. Instead of producing bilayers as thin surfaces on the XY plane, the printed axis of bilayers is rotated to allow the design and fabrication of differentiated “bilayer” structures. It facilitates the variation in mechanical and hygroscopic properties within the depth layers of the cross-sectional structures. Thus, the developed method enables the generation of differentiated monomaterial systems with considerable shape transformation. The depth of the self-shaping structure can be significantly increased while maintaining the speed and amount of actuation. Four dimensionally printed self-shaping structures can therefore be upscaled from 1 mm to 58.7 mm or more, even with commercially available desktop printers. Using this method, one can create hierarchical structures on a larger scale and with a precision closer to biological materials. The proposed CS4DP approach broadens the design space of bioinspired motion mechanisms to high-resolution, porous, and upscaled self-shaping structures. The fundamental investigations will be further expanded to large-scale additive manufacturing platforms, which leads the way for a range of possible building-scale applications of CS4DP structures.

## Figures and Tables

**Figure 1 biomimetics-08-00233-f001:**
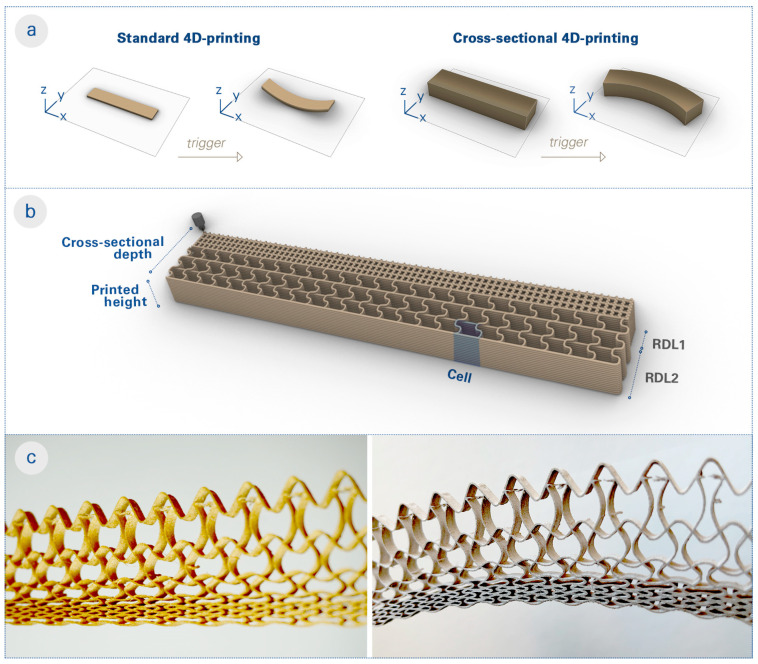
Conceptual explanation of cross-sectional 4D-printing (CS4DP). (**a**) Diagram showing the difference between standard 4D-printing and CS4DP, which entails rotating the printed axis of bilayers and underlines the monomateriality and scalability of the proposed method. (**b**) Illustration of a CS4DP structure composed of multiple RDLs with tunable mechanical and hygroscopic properties through the functional patterning. (**c**) Photos of a specimen fabricated with the CS4DP method, before (**left**) and after (**right**) actuation, which highlight the differentiated cell resolution and wall geometry in its cross section. Upon the actuation trigger (water submersion and subsequent drying), the differentiated dimensional changes of cells within the cross section cause the CS4DP structure to self-shape to the programmed curvature. Unlike standard 4D-printing, CS4DP can achieve large deformations with large cross-sectional depth and as a monomaterial system.

**Figure 2 biomimetics-08-00233-f002:**
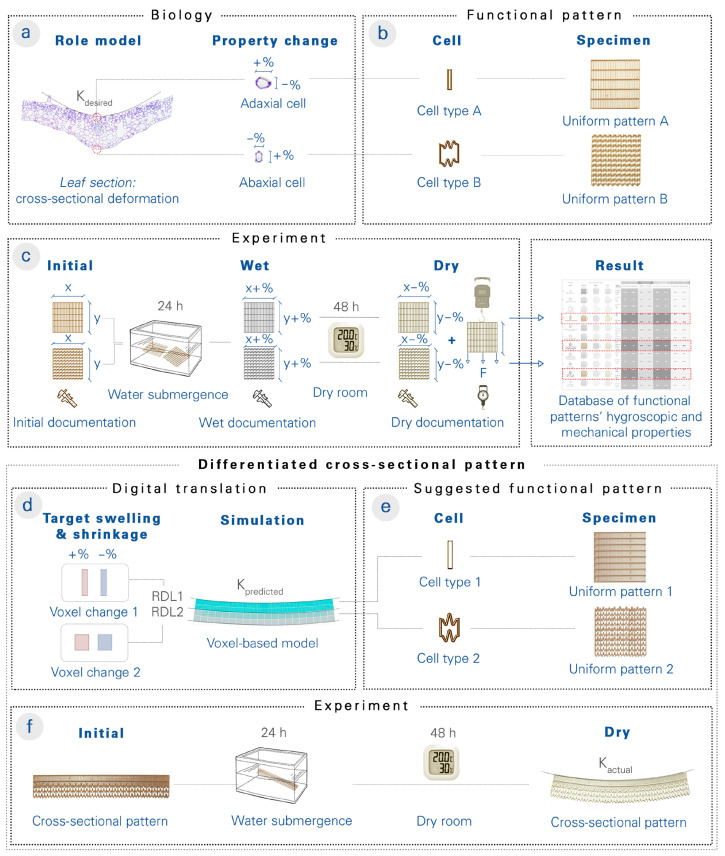
The workflow for testing the hypotheses of biological role models using CS4DP. (**a**) From cross-sectional imaging of the large-flowered butterwort (*Pinguicula grandiflora*), cellular dimensions of the adaxial (**upper**) and abaxial (**lower**) epidermis are measured before and after its depression formation. (**b**) The cellular changes are translated into functional patterns, and uniform test specimens are 4D-printed. (**c**) The uniform test specimens are measured before, during, and after wetting, and the physical swelling and shrinkage data, as well as the ultimate load and deformation data, are entered into a database. (**d**) To design cross-sectional structures with desired curvatures, target swelling and shrinkage values for RDL cells are iteratively defined and simulated with the help of the voxel-based model before 4D-printing. (**e**) Through the experimental results of the uniform test specimens, the RDLs’ functional pattern is suggested based on the observed correlation between hygroscopic deformation and functional pattern parameters. (**f**) After actuation triggering, the curvature of the actual, physical specimen is compared to the digital, predicted curvature, and the results are used to improve the extensible database of functional patterns.

**Figure 3 biomimetics-08-00233-f003:**
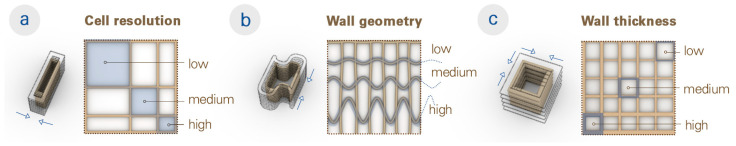
Functional pattern parameters used to tailor the amount and direction of shrinkage and swelling in each cell of the overall CS4DP structure. (**a**) The cell resolution refers to the size of the individual cells. (**b**) The wall geometry is controlled by the printed tool path. Higher amplitudes add more length to the cell wall, while lower amplitudes have less added length. (**c**) The wall thickness is controlled by the flow of material deposition when printing the cell walls.

**Figure 4 biomimetics-08-00233-f004:**
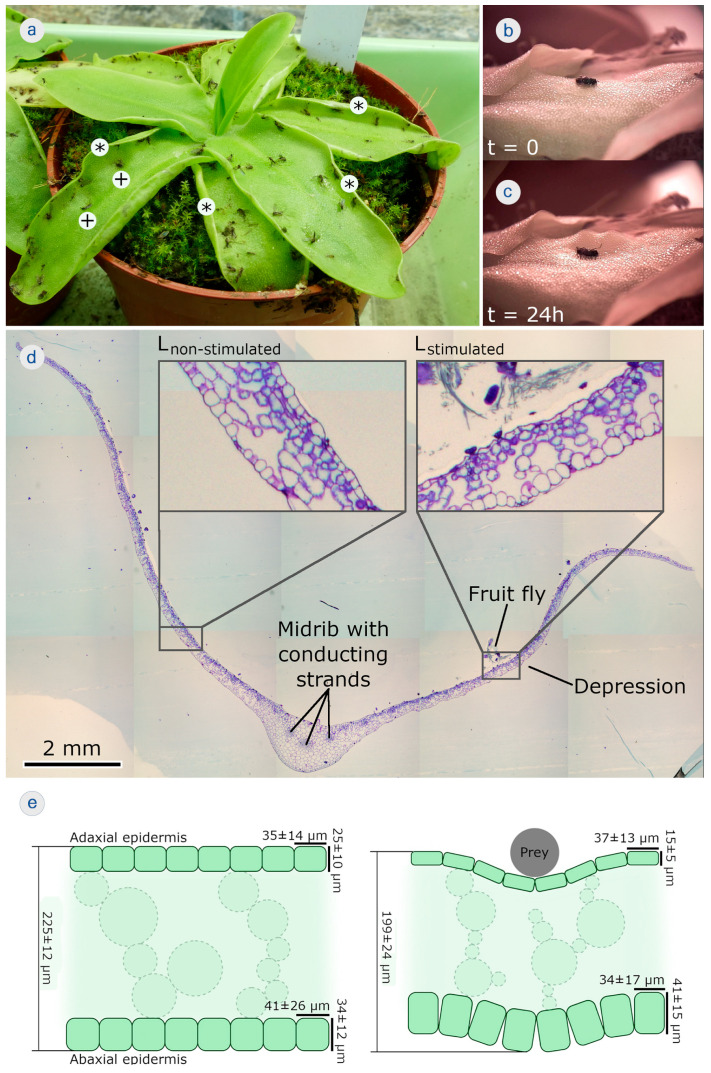
Trap leaf motions in the carnivorous large-flowered butterwort (*Pinguicula grandiflora*). (**a**) A cultivated specimen. The plant catches a variety of prey, mainly small fungus gnats, in culture all by itself. Depressions (+) and rolled leaf margins (*) are indicated. (**b**) Feeding with a fruit fly, leading to (**c**) the local formation of a depression on the trap leaf, which is completed after 24 h. (**d**) Light microscopical image with toluidine staining of a semithin cross section of a trap leaf. The midrib and conducting strands are indicated. Placing a prey animal (fruit fly) onto the right half of the leaf entailed formation of a depression, whereas the left half of the leaf did not respond. The insets are detailed images of the non-stimulated (without depression) and of the stimulated leaf regions (with depression). Remnants of the partly digested fruit fly are still visible. (**e**) Schematic cross section of a leaf segment before stimulation by prey (**left**). The epidermises with mean ± SD values of cellular dimensions, as measured, are indicated. The same leaf fragment is then stimulated by prey (**right**). A depression has formed. The mean ± SD values of cellular dimensions, as measured, are indicated.

**Figure 5 biomimetics-08-00233-f005:**
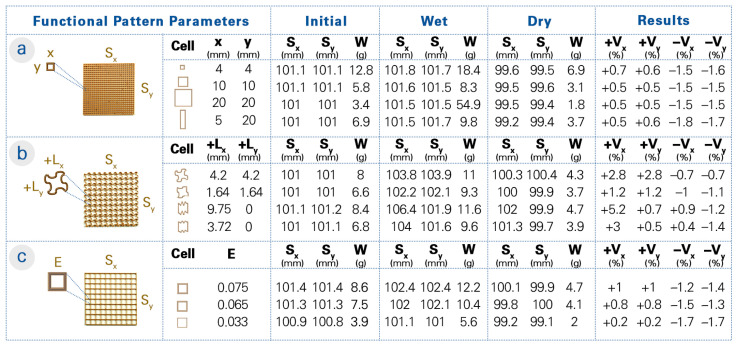
A parameter study on the hygroscopic properties of uniform functional patterns. (**a**) The resolution is controlled through the cell size (x and y in mm). (**b**) The wall geometry is controlled through the curvature of the extrusion path, which results in an added length (+L_y_ and +L_y_ in mm). The size (S_x_ and S_y_ in mm) and weight (W in g) of the 4D-printed test specimens were measured after printing, after 24 h water submergence, and after 48 h drying in 30% RH. The resulting shrinkage and swelling data (+V_x_, +V_y_, −V_x_, and −V_y_ as a percentage) was then correlated to the digital voxel-based model. (**c**) The wall thickness is controlled through the amount of material extrusion (flow rate E).

**Figure 6 biomimetics-08-00233-f006:**
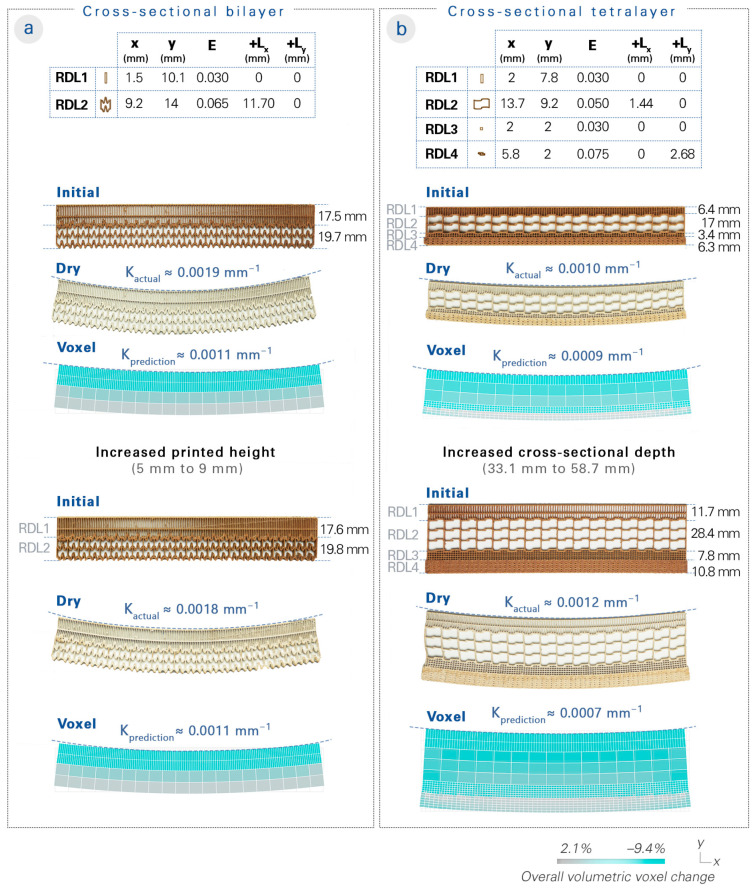
Cross-sectional structures inspired by the large-flowered butterwort. Two options for the biological hypothesis were tested. The cross-sectional bilayer comprises two RDLs (representing the upper and lower epidermis), while the cross-sectional tetralayer comprises four RDLs (representing the upper epidermis, upper mesophyll, lower mesophyll, and lower epidermis). The cell sizes (x and y in mm), wall thicknesses (flow rate E), and added lengths from the wall geometry (+L_x_ and +L_y_ in mm) are listed. The two options of cross sections with distinct depth layers were then 4D-printed. After the actuation triggering and shape deformation, the curvatures of the test specimens were measured. (**a**) The cross-sectional bilayer resulted in a curvature of approximately 0.0019 mm^−1^, compared to the voxel-based model prediction of 0.0011 mm^−1^; an additional test specimen was produced to show that the curvature could be maintained even with an increase in printed height (from 5 to 9 mm). (**b**) The cross-sectional tetralayer resulted in a curvature of about 0.0010 mm^−1^, compared to the voxel-based model prediction of 0.0009 mm^−1^; an additional test specimen was produced with increased cross-sectional depth (from 33.1 to 58.7 mm), which even increased the curvature.

## Data Availability

The data presented in this study are contained within the article or the Appendix A.

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
