# Peer review of "Cross-Sectional 4D-Printing: Upscaling Self-Shaping Structures with Differentiated Material Properties Inspired by the Large-Flowered Butterwort (Pinguicula grandiflora)"

_biomimetics, 2023, doi:10.3390/biomimetics8020233_

Round 1

Reviewer 1 Report

This manuscript presents a very interesting work regarding cross-sectional 4D printing. The bio-inspired idea is relatively novel and the research design is appropriate. The results have great significance for the development of 4D printing. It is suggested that the manuscript can be considered for publication.

Author Response

We express our gratitude for reviewing our manuscript. We greatly appreciate that you have recognized the significance and contribution of our work, and we thank you for suggesting it for publication. 

Reviewer 2 Report

This is an interesting paper describing upscaling self-shaping structures with differentiated material properties inspired by the large flowered Butterwort. Using this method, one can create hierarchical structures on a larger scale and with precision closer to biological materials. This study is relatively innovative. Before the publication in Biomimetics, some issues are necessary to be solved as the following comments suggest:

1) In the introduction section, more background should be added. Some examples of existing 4D printing should be introduced, and more references are needed to support the background.

2) The structure of the introduction section might be better to change. There is no need to break it into three parts.

3) The first abbreviation of “Cross-sectional 4D printing” should appear in the text instead of in the annotation of Figure 1.

4) During the deformation process, does the multi-layered structure appear stratification phenomenon? It's best to explain it in the paper.

5) How long will the multilayer material last? And in this study, the elastic modulus, ductility and strength of the multi-layered structure were not tested.

6) As for the deformation shown in Figure 5, can simulation of surface stress be added to support it?

7) Consider whether CS4DP printing can be divided into ordered, quasi-ordered and unordered levels? If so, the slip phenomenon in the ordered mode needs to be considered. It is necessary to analyze the slip under different printing parameters, such as the effect of printing speed on the slip distance.

8) Multi-layered structure made by CS4DP printing, what materials are used?

9) The word size of serial number in the picture is too small, which should be larger than word in figure. The clarity of the picture should be improved.

Moderate editing of English language

Reviewer 3 Report

The work is very interesting, the manuscript is clear, and the work presented goes beyond the previous ones and is scientifically well written, to deserve publication after some changes that I address in the next paragraph.

 1.       In topic 1.3 please improve the manuscript in order to clarify the uniqueness of the cross-sectional 4 D printing compared with the “regular” 4D printing. Figure 1 does not help to understand the novelty of the process.

2.       Caption of Figure 1 should be more clear. In a) it should be mentioned how did the drawing was obtained. I can’t understand if c) is a photo of the printed structure or not. The results presented here were not mentioned yet in the text of the manuscript, maybe this figure should appear later in the text.

3.       Topic 2.2  - “a voxel-based model based on constraints-based projection modeling”, please avoid repletion of the word based.

4.       Topic 2.3 how many replicas of the 4D printed specimens were produced and measured?

5.       Figure 3 – use a different symbol for depression and rolled leaf margins, since it can confuse the reader, maybe numbers. There is no image for “(f) The same leaf fragment after stimulation by prey. A depression has formed. The mean ± SD values of cellular dimensions, as measured, are indicated.“, please verify the correspondence of caption – figure presented.

6.       Topic 3.1 – “Lstimulated shows clear differences in tissue and cellular dimensions when compared to Lnon-stimulated. The overall mean ± SD leaf thickness is 225 ± 12 μm (n=10) for Lnon-stimulated and 199 ± 24 μm (n=11) for Lstimulated. Adaxial epidermal cells analyzed in Lstimulated in the vicinity of the depression are 37 ± 13 μm (n=69) wide and 15 ± 5 μm (n=69) high, as seen in the cross section, and those for Lnon-stimulated in a comparable area (i.e., mirrored about the mid-rib) are 35 ± 14 μm (n=76) wide and 25 ± 10 μm (n=76) high. Generally, the mean thickness of the adaxial epidermis in Lstimulated in the vicinity of the depression accounts for only ~60% of the mean thickness of the epidermal layer in Lnon-stimulated in a comparable area. Whereas the abaxial epidermis in the Lnon-stimulated region (comparable in size to the stimulated one on the other leaf side) is characterized by cells with average widths of 41 ± 26 μm (n=71) and heights of 34 ± 12 μm (n=72), cells in the Lstimulated region are 34 ± 17 μm (n=96) wide and 41 ± 15 μm (n=99) high. In summary, whereas the mean abaxial epidermal cell widths in Lstimulated are ~17% smaller, the mean heights are ~21% greater than in Lnon-stimulated.” Should be presented in a table.

7.       Section 4 - In order to state “the developed CS4DP technique results in differentiated and layered structures with varied mechanical and hygroscopic properties in the cross section” there should be present in the study some mechanical tests to support this.

8.       Please clarify how was performed the rotation of the printing axis. This information should be more clear in the manuscript.

9.       References: The document could have a larger number of references, include references regarding 3D and 4D printing, nature inspired active materials for example: Tibbits, Skylar, et al. "4D printing and universal transformation." (2014).;  Raviv, Dan, et al. "Active printed materials for complex self-evolving deformations." Scientific reports 4.1 (2014): 1-8. ; Tibbits, S. The emergence of “4D printing”. In Proceedings of TED Conferences. Available online: https://youtu.be/0gMCZFHv9v8 ; “Nature-Inspired Cellulose-Based Active Materials: From 2D to 4D. Appl. Biosci. 20232, 94-114. https://doi.org/10.3390/applbiosci2010009.”,

The English is clear and only some minor changes need to be addressed. 

Round 2

Reviewer 2 Report

The authors have addressed my issues.